# Emerging Roles for DNA 6mA and RNA m6A Methylation in Mammalian Genome

**DOI:** 10.3390/ijms241813897

**Published:** 2023-09-09

**Authors:** Leijie Xie, Xiaosong Zhang, Jiaxiang Xie, Yanru Xu, Xiao-Jiang Li, Li Lin

**Affiliations:** Guangdong Key Laboratory of Non-Human Primate Research, Laboratory of CNS Regeneration (Ministry of Education), Guangdong-Hongkong-Macau Institute of CNS Regeneration, Jinan University, Guangzhou 510632, China; xieleijie@stu2021.jnu.edu.cn (L.X.); zxs2569778031@163.com (X.Z.); tsegaaicoeng@hotmail.com (J.X.); yanruxu202102@163.com (Y.X.); xjli33@jnu.edu.cn (X.-J.L.)

**Keywords:** methylation, N6-methyldeoxyadenosine, N6-methyladenosine, mammalian genome

## Abstract

Epigenetic methylation has been shown to play an important role in transcriptional regulation and disease pathogenesis. Recent advancements in detection techniques have identified DNA N6-methyldeoxyadenosine (6mA) and RNA N6-methyladenosine (m6A) as methylation modifications at the sixth position of adenine in DNA and RNA, respectively. While the distributions and functions of 6mA and m6A have been extensively studied in prokaryotes, their roles in the mammalian brain, where they are enriched, are still not fully understood. In this review, we provide a comprehensive summary of the current research progress on 6mA and m6A, as well as their associated writers, erasers, and readers at both DNA and RNA levels. Specifically, we focus on the potential roles of 6mA and m6A in the fundamental biological pathways of the mammalian genome and highlight the significant regulatory functions of 6mA in neurodegenerative diseases.

## 1. Introduction

Epigenetics aims to explore the molecular mechanisms for stable genetic modifications of gene expression, protein functions, and ultimately cell fate without altering the DNA sequence, including DNA modification, RNA modification, histone modification, and non-coding RNA. These mechanisms regulate gene expression and chromatin structure. Recent studies have revealed the methylation on N-6 of adenine is a novel epigenetic modification that can be found at adenosine of DNA (N6-methyldeoxyadenosine, 6mA) and RNA (N6-methyladenosine, m6A). Both 6mA and m6A are present in mammalian cells, and notably, the m6A is abundant in the brain than in other mammalian tissues [1]. For DNA methylation, 5-methylcytosine (5mC) at CpG dinucleotides is the most abundant modified DNA base [2,3,4,5]. In addition, 5-hydroxymethylcytosine (5hmC) is not only an intermediate product of DNA demethylation but also a novel stable epigenetic regulator. Imbalances in the 5hmC levels contribute to neurological diseases, including neurodevelopmental and neurodegenerative diseases [6]. Similarly, 6mA, as a novel epigenetic modification on the DNA level, is highly enriched in the adult brain, regulates learning gene expression, and is associated with fear formation [7]. However, several studies have shown that the abundance DNA 6mA in the mammalian and even the human genome is very low [6]. A currently controversial issue is the presence of DNA-6mA in mammalian genomes [8,9].

Different RNA modifications have been reported on messenger RNA (mRNA), transfer RNA (tRNA), ribosomal RNA (rRNA), microRNA (miRNA), small nuclear RNA (snRNA), and long non-coding RNA (lncRNA) [10,11,12]. m6A is the most abundant in eukaryotic cells. Substantial evidence indicated that m6A was enriched in mammalian brains and embryonic stem cells (ESCs) and that its abundance increased continuously from the embryonic to the adult brain [1,13]. These results implicated that m6A may be involved in brain development, functional regulation, and synaptic plasticity. It has been demonstrated that m6A methylation affected neural stem cells, learning and memory, brain development, axon growth, and glioblastoma [14]. Abnormal m6A may cause neurological development and mental illness [15].

DNA 6mA has been reported early in prokaryotes and was not identified in eukaryotes until the last decade. Genome-wide profiling implicated a role for 6mA in regulating gene expression [16]. In contrast, RNA m6A methylation was studied earlier and widely. Increasing evidence uncovered the important role of m6A in human cancer progression and tumorigenesis [17]. In recent years, numerous studies have reported the genomic distribution and dynamics of 6mA and m6A in mammals, as well as their underlying functions in the regulation of gene expression. The higher abundance of 6mA and m6A in the mammalian brain sparked our interests in understanding their potential role in the central nervous system (CNS), specifically in development- and aging-related diseases. Current studies have reported a potential relationship between m6A and aging and neurodegenerative diseases; however, little is known about the role of 6mA. Interestingly, the methyl group is added to the same position of “A” at both DNA and RNA. We speculated that 6mA and m6A may somehow participate in the same physiological process in organisms somehow, either coordinated or independently. Evidence came from our unpublished data showing 6mA were accumulated in the brain of monkeys with aging, suggesting that 6mA exerted an epigenetic regulatory role in normal aging. Here, we focus on molecular mechanisms of 6mA and m6A in the biological pathways. Finally, we discuss the important role of 6mA in neurodegenerative diseases.

## 2. DNA 6mA Methylation and Its Writers, Erasers, and Readers

DNA 6mA was first discovered in prokaryotes and it widely exists in many species, which involves in the regulation of DNA replication, repair, transcription, transposition, and cell defense. It is produced by the addition of the methyl group from S-adenosyl-l-methionine (SAM) to N-6 of the adenine ring via specific methyltransferases [3]. In recent years, it has been reported that 6mA exists in eukaryotes, such as green algae [18], *C. elegans* [19], and *Drosophila* [20], but 6mA has been proved to exist at extremely low levels in most higher eukaryotes, especially in mammals.

DNA 6mA modification is a dynamic process including the addition and removal of methyl groups, coordinated via a complex system of methylases. These proteins are called writers and erasers. In eukaryotes, several methyltransferases have been reported (Figure 1) [21]. DNA N6 adenine methyltransferase 1 (DAMT-1) has been identified in *C. elegans* to methylate 6mA via the knock down and overexpression of treatments. The DNA methyltransferases (DNMTs) family is known to act as 5mC methyltransferases in animals [22]. The overexpression or depletion of DNMT1 lead to the increased or decreased levels of genomic 6mA [23]. This suggested that DNMT1 may be a 6mA methyltransferase [24]. Further analysis revealed that DNMT1 belongs to the MT-A70 domain-containing enzyme family which includes mRNA methyltransferases Ime4 and Kar4 in yeast, and the methyltransferase-like protein 3 (METTL3) and METL14 in humans [25,26,27]. In mammals, METTL4 is a DNMT1 homologue and a paralogue of METTL3 and the methyltransferase-like protein14 (METTL14), which functions as a DNA 6mA methyltransferase. In the human genome, N-6 adenine-specific DNA methyltransferase 1 (N6AMT1) has reported as a methyltransferase, which contains the catalytic conserved motif NPPY [28]. This conclusion has been demonstrated via structural analysis and silence/overexpression.

6mA modification is a reversible process, and there must be a demethylation process. 6mA demethylase (DMAD), as a ten-eleven translocation (TET) homologue in *Drosophila* [20], tightly controls the demethylation of 6mA during embryonic development and tissue homeostasis. Loss of DMAD may result in severe developmental defects, while increasing the abundance of 6mA modifications. Interestingly, DMAD is expressed at higher levels in the *Drosophila* brain than in the ovary. When DMAD was depleted, the levels of 6mA in the brain increased 100-fold [29]. The ALKB protein family catalyzes the demethylation of various methylated DNA and RNA nucleotides. N6-methyladenine demethylase 1 (NMAD-1) is a member of the ALKB family. The deletion of *Nmad-1* in *C. elegans* cells resulted in increased 6mA levels [30]. Therefore, it can be stated that NMAD-1 is a 6mA demethylase. Another member of the ALKBHs (alkB homologs), ALKBH1, also showed to mediate the demethylation of 6mA in mouse and human cells [31]. The 6mA levels were higher in glioblastoma cells compared with normal tissues, and the 6mA demethylase ALKBH1 has also been considered as a potential new target for cancer treatment [32]. Further, it has been confirmed that the human fat mass and obesity-associated protein (FTO) not only is a demethylase of RNA m6A, but also demethylates DNA 6mA. In human cancer tissues, 6mA showed a downward trend, accompanied by the downregulation of N6AMT1 and the upregulation of ALKBH1. Therefore, the downregulation of genomic 6mA levels may promote tumorigenesis [28].

It is important to note that the function of 6mA requires binding proteins in addition to methyltransferases and demethylases. These binding proteins, called readers, can recognize and bind 6mA sites on DNA and play a significant role in gene expression regulation and cell function [33]. Protein readers, YTHDC5, YTHDF1, and YTHDF2, have been identified as bona fide 6mA regulators in type II testicular germ cell tumors (GCT) via reanalysis of the expression microarray data [34]. Additionally, HNRNPC, which is highly expressed in GCT tissues and cell lines, was an “indirect reader” involved in the “6mA switch” that altered the local RNA structure, thereby triggering the RNA-binding motif and facilitating protein binding [34]. In human mitochondria, the single-stranded DNA-binding protein 1 (SSBP1), containing the HNRNP and YTH domains, was thought to be a protein that recognized 6mA, and it preferentially bound to ssDNA along the heavy chain, consistent with the location of 6mA enrichment [35,36]. The Jumu protein was an important transcription factor in *Drosophila*, encoding 719 amino acids. It was the first FOX family protein with dual functions discovered, which could cause the large-scale modification of heterochromatin and has a specific recognition function. The Jumu protein has been reported to recognize and bind 6mA-modified DNA and act as a maternal factor to control the proper activation of the zygotic genome. Zelda, an essential precursor of the transcription factor for the maternal to zygotic transition (MZT), was one of the key target genes of Jumu. The study demonstrated that Jumu proteins regulated zygotic genome activation, at least in part by recognizing Zelda that controlled the 6mA mark, thereby regulating early embryogenesis [37].

These studies help us understand the related proteins of 6mA modification in cells, and further structural and biochemical investigations are needed to find the methyltransferases, demethylases, and binding proteins of 6mA in mammals.

## 3. The Potential Functions of DNA 6mA Methylation

The 6mA modification has been well described in prokaryotes to play roles in foreign DNA cleavage, virus defense, and DNA damage repair [38,39,40,41]. The potential function of 6mA modification in mammals remains an active area of ongoing research. In addition to providing binding sites for effector proteins, one potential effect of the 6mA modification was to directly alter the overall structure of DNA. Early crystal structures suggested that 6mA may also alter DNA secondary structure [42]. Meanwhile, 6mA affected DNA double helix formation by altering the base pair stability and base stacking [40,42]. Similar to the main function of 5mC, 6mA modification has been found to play a role in regulating gene expression in different species [43]. The 6mA modification reduced the thermal stability of the DNA chain, which in turn changed the curvature of the DNA chain; thus, 6mA at a specific position affected the interaction of transcription factor-related elements [44]. 6mA modification acted as an epigenetic mark that affected the chromatin structure and gene expression patterns in a heritable manner. Recent studies showed that the coordination of 6mA and histone modification H3K4me2 contributed to the transgenerational epigenetic control [19], and 6mA was mainly present at ApT dinucleotides around the transcription start site (TSS) with a bimodal distribution and appeared to mark the active genes in Chlamydomonas. Furthermore, a genome-wide map of 6mA and its genomic distribution suggested a potential epigenetic role for 6mA in regulating gene expression [18]. Together, these findings suggested 6mA functions as an epigenetic mark in eukaryotes.

In mouse embryonic stem cells, 6mA was associated with the repression and silencing of genes, particularly those on the X chromosome, and was known to play an important role in cell fate decisions. During the first 120 h of zebrafish embryonic development, the 6mA levels rose steadily from a pluripotent cell to a nearly fully formed individual, and the same pattern was observed in mouse embryonic days 7–21 [45]. With the continuous improvement of detection technology, 6mA modification has been detected in humans and is confirmed to be related to various diseases. It has been reported that 6mA was involved in the pathogenesis of essential hypertension [46]. The 6mA levels in leukocytes of all hypertension models were significantly reduced and could return to normal levels after successful hypertension treatment. The decreased 6mA levels in leukocytes and vascular smooth muscle cells (VSMCs) of the hypertensive models in vivo and in vitro were due to the increased ALKBH1 content. The 6mA was a sensitive marker for the development, diagnosis, and treatment of hypertension [46]. Moreover, 6mA was involved in the regulation of fatty liver degeneration. In patients with fatty liver degeneration, significantly increased the 6mA levels and downregulated ALKBH1 have been found [47]. Integrative analysis of transcriptome and chromatin immunoprecipitation-sequencing revealed that ALKBH1 directly bound and specifically demethylated the 6mA of genes involved in fatty acid uptake and lipogenesis, thereby reducing hepatic lipid accumulation [47]. Importantly, ALKBH1 overexpression was sufficient to suppress lipid uptake and synthesis and attenuate diet-induced hepatic steatosis and insulin resistance. ALKBH1-induced 6mA played an integral role in hepatic fatty acid metabolism as an epigenetic repressor and provided a potential therapeutic target for treatment.

Aberrant DNA methylation has been shown to be associated with tumorigenesis. Recent reports indicated that 6mA participated in the progression and tumorigenesis of several cancers, such as breast cancer [48], gastric cancer [49], liver cancer [28], lung cancer, [50] and glioma [32]. Reduction in N6AMT1 in clinical breast cancer tissues correlated with 6mA intensity and predicted poorer survival of patients [48]. Functionally, the knockdown of N6AMT1 significantly decreased the 6mA levels in DNA and promoted colony formation and the migration of breast cancer cells, whereas the overexpression of N6AMT1 had the opposite effect. Mechanistically, N6AMT1 acted as a methyltransferase for 6mA formation and repressed the gene expression of key cell cycle inhibitors such as RB1 and TP53, which was also the first study on the regulation and function of 6mA modification in breast cancer progression and prognosis.

The 6mA levels were apparently tissue specific, with the highest 6mA levels observed in the brain. It is well known that environmental exposure may induce epigenetic changes. What need to be pointed out here is that environmental factors in vivo and in vitro are inseparable from the expression level of 6mA. Yao et al. demonstrated that 6mA in the mouse brain was a dynamic change in response to environmental stress, and the change in 6mA was related to the expression of neuronal genes and LINE transposon. Stress-induced 6mA changes significantly related to genes which are associated with neuropsychiatric disorders. These results implicated that 6mA may play a key epigenetic role in mammalian brain development and pathogenesis [51]. Substantial evidence showed that physiological and psychological stress can alter DNA methylation of key stress-related genes in the mouse brain [52,53]. The prefrontal cortex (PFC), responsible for the highest-order cognitive abilities, is particularly vulnerable to chronic stress and plays an important role in depression. It has also been reported that after characterizing the dynamics of 6mA under starvation in the single-cell model organism, Tetrahymena thermophila, single-molecule real-time sequencing (SMRT-sequencing) showed that the 6mA levels in starved cells were significantly lower compared with the DNA 6mA levels in vegetatively growing cells [54]. In summary, environmental factors are extremely important for the influence of 6mA in different model organisms.

6mA is emerging as a crucial DNA epigenetic mark to regulate gene expression in eukaryotes. Current researchers explored the role of 6mA dysregulation and 6mA modulators in human diseases, especially in cancers [55]. However, the study of 6mA in mammals is still in its infancy, so further studies are needed to confirm the characterization of 6mA in mammals and expand our knowledge of the biological functions of 6mA in human diseases, especially the physiological significance of its high abundance in the mammalian brain.

## 4. RNA m6A Methylation and Its Writers, Erasers, and Readers

m6A is not only present in mRNAs, but also abundantly present in non-coding RNAs, including rRNA, miRNA, snRNA, circRNA, snoRNA, and lncRNAs [56,57,58]. Recent sequencing-based technological breakthroughs have enabled the genome-wide profiling of m6A [59,60,61]. In mammals, m6A is widely distributed in many tissues, and the abundance of m6A exhibits cell line-, tissue- and organ-specificity. Notably, the brain shows the highest m6A abundance among mammalian organs [1]. The m6A not only has the characteristics of temporal and spatial distribution, but also has evolutionary conservation in the mammalian brain [1], implying that m6A modification on RNAs may play an important role in the function and regulatory network of mammalian CNS, such as synaptic plasticity, axon regeneration, learning, memory, and cognition [62,63,64,65,66]. The presence of m6A has been detected in several species including *E. coli* [67], yeast [68,69], *Arabidopsis thaliana* [70,71], zebrafish [72,73], *Drosophila* [74,75], mice, and humans{Citation}. m6A sites are highly conserved, usually rich in RRACH consensus motifs (R = G or A and H = A, C or U), and m6A is mainly located near the mRNA 3′ untranslated regions (3′UTRs) and the stop codon of the protein coding sequence [76].

At the molecular level, m6A regulates RNA metabolism, including mRNA, rRNA, tRNA, miRNA, and circRNA, especially the splicing, stability, localization, and translation of mRNAs, which plays important roles in differentiation, development, and metabolism [77,78,79,80,81,82]. m6A has emerged as a key regulator of many important biological processes in physiology and diseases. Meyer et al. found that the distribution of brain m6A changes in genes accounted for about 94.8%, of which the proportions of the protein coding regions (CDS), untranslated regions (UTRs), and introns were 50.9%, 41.9%, and 2.0%, respectively [1].

The methylation of m6A is a dynamic and reversible process, which is jointly regulated by the methyltransferase (writers), demethylase (erasers), and binding proteins (readers). The writers deposit m6A on a specific subset of RNAs, and the erasers can remove the m6A marker from RNAs. As an epigenetic marker, m6A alters the RNA secondary structure and attracts/repels specific binding proteins, which further regulates gene expression via RNA processing, localization, translation, and degradation. The readers are attracted by m6A on RNAs and drive downstream functions to specific sites in the RNA (Figure 2).

METTL3, METTL14, and Wilms’ tumor 1-associating protein (WTAP) are essential components of mammalian m6A methyltransferases, and other constituent proteins of the methyltransferase complex include METTL5, tRNA methyltransferase activator subunit 11-2(TRMT112), Cbl proto-oncogene like 1 (CBLL1), zinc finger protein 217 (ZFP217), RNA binding motif protein 15 (RBM15), RNA binding motif protein 15B (RBM15B), vir-like m6A methyltransferase-associated protein (VIRMA), zinc finger CCCH-type containing 13 (ZC3H13), and methyltransferase-like protein 6 (METTL16). Abnormalities in the function or expression of these writers led to disordered m6A regulation and affected mRNA metabolism and gene expression. METTL3 is an S-adenosylmethionine-dependent methyltransferase that specifically recognizes the conserved RRACH motif on RNAs [83]. METTL14 can combine with METTL3 to form a stable heterodimer, significantly enhance the catalytic activity, and promote the recognition of RNA methylation sites [84]. The knockdown of METTL3 and METTL14 by siRNA/shRNA in mESCs resulted in an dramatic decrease in the m6A levels and regulated pluripotency maintenance and departure by affecting the pAkt and pErk signaling pathways [85]. ZFP217 interacted with METTL3, leading to a decrease in the global m6A levels, which affected ESCs’ self-renewal and the somatic cells’ reprogramming [86]. METTL3-knockout in ESCs resulted in a m6A decrease and reduced the key naive pluripotency-promoting transcripts [87]. These studies revealed important functions of m6A writers in cell pluripotency and reprogramming. The deletion of *Mettl3* caused a decrease in the 6mA and Ezh2 levels in mice and inhibited the proliferation and neuronal development of adult neural stem cells (aNSCs), whereas the overexpression of histone methyltransferase Ezh2 reversed this effect [88]. The high expression of METTL3 in the mouse hippocampus could significantly enhance the consolidation of long-term memory in mice. Sustained depletion of METTL3 in the hippocampus reduced the ability of mice to form long-term memories, but interestingly, it did not alter the ultimate learning effect if trained sufficiently [65]. This may be due to the immediate early genes (IEGs), which is an early gene that combined the formation of long-term memory with the expression of short-term memory [89,90]. The insufficient expression of IEG proteins may lead to synaptic plasticity or a disorder of long-term memory function [91]. Interestingly, the synthesis of IEG proteins was dependent on METTL3 [92]. WTAP may act as a regulatory subunit in the m6A methyltransferase complex, and also target RNA and recruit the complex to RNA. The knockdown of WTAP inhibited the catalytic activity of the METTL3-METTL14 complex [93]. The loss of WTAP resulted in decreased m6A levels and cerebellar Purkinje cell damage in mice [94].

FTO and alkB homolog 5 (ALKBH5) has been identified as m6A demethylases in eukaryotes [95]. Subsequent studies showed that FTO can stepwise oxidize m6A to two previously unknown intermediates, N6-hydroxymethyladenosine (hm6A) and N6-formyladenosine (f6A) [96]. Differently, ALKBH5 directly reversed m6A to adenosine, with no intermediate detected [97]. FTO exhibited high expression levels in the brain, especially in neurons [98]. The inhibition of FTO via the rhein/siRNA treatment led to an accumulation of m6A modification, a reduced local translation of growth associated protein 43 (GAP-43) mRNA, and a repressed axon elongation [99]. FTO overexpression upregulated the level of ephin-B2 through an m6A-YTHDF2-dependent manner and reversed the downregulation of ephin-B2 in the manganese-induced parkinsonism model [100]. Several studies also showed that FTO was involved in neurogenesis, differentiation, learning, memory, and cognition processes in CNS [101,102,103]. ALKBH5 was enriched in neurons and played an important regulatory role in biological processes such as mRNA processing [104] and short-term plasticity process [105]. Furthermore, ALKBH5 was enriched in diploid primary spermatocytes and the loss of ALKBH5 in mice resulted in impaired fertility and aberrant apoptosis [106,107].

The reader proteins are performers of the specific regulatory function of m6A modification. Several protein families have been identified that retain the ability to recognize and bind m6A modification, including the YTH domain family (YTHDF1-3), YTH domain-containing proteins (YTHDC1-2), heterogeneous nuclear ribonucleoproteins (HNRNPC/G, HNRNPA2B1), insulin-like growth factor 2 mRNA-binding proteins (IGF2BP1-3), eukaryotic initiation factor 3 (eIF3), proline-rich coiled-coil 2A (PRRC2A), and Fragile X mental retardation protein (FMRP). Mammalian YTHDF proteins include three members, YTHDF1, YTHDF2, and YTHDF3, which act as readers in the cytoplasm to regulate m6A degradation and translation. YTHDF1 can interact with eukaryotic translation initiation factor eIF3 and recruit it to mRNA to improve the translation efficiency [108]. YTHDF2 can selectively bind to m6A-modified mRNAs and transport them to decay sites to regulate the RNA lifespan [80]. YTHDF3 cooperates with YTHDF1 to interact with 40S/60S ribosomal subunits, ultimately enhancing the mRNA translation of mRNA [109]. Interestingly, there may be a significant coordination and compensation of YTHDF1-3, where the three proteins jointly regulate the mRNA metabolism [110,111]. The YTHDC proteins (YTHDC1, YTHDC2) are another important class of m6A readers. YTHDC primarily functions in RNA splicing and nuclear localization [77,112]. YTHDC2 enhanced the translation efficiency by resolving the secondary structures of the structured mRNA [113,114]. In recent years, a large number of m6A readers and their partners have been discovered, and their connections with RNA metabolism, physiology, and diseases have also been continuously revealed. Interestingly, m6A repels proteins that have been reported as a new class of m6A–protein interactions [115].

## 5. The Potential Functions of RNA m6A Methylation

RNA methylation is an important part of epigenetic modification, which regulates the structural properties of RNA, as well as affects the affinity between RNA and ribosomes. Compared to DNA 6mA methylation, the function of RNA m6A methylation has been extensively studied. In particular, roles for m6A methylation, demethylation, and the enzyme-mediated regulation of target RNA have been confirmed. The functions of m6A are tightly associated with its writers, erasers, and readers. The mutations or abnormal expression of these writers/erasers/readers in the CNS may lead to the dysregulation of the epitranscriptome, resulting in physiological and pathological changes [64,66,116,117].

m6A acted as a marker signal that can be recognized by m6A reader proteins for specific splicing kinetics [118]. In 2012, the first transcriptome-wide analysis of m6A sites in the human and the mouse using methylated RNA immunoprecipitation sequencing (MeRIP-seq) [76] showed that m6A methylation is often enriched in the vicinity of introns and stop codons. The colocalization of METTL3 and the splicing factors provided spatial possibility that m6A played a role in splicing to silence the expression of METTL3 in HepG2 cells. These results suggested that m6A played an important role in regulating the splicing events of multiple heterogeneous genes as well as differentiating exons and introns [93,119]. In addition, human m6A could regulate the selective cleavage of MyD88 in response to lipopolysaccharide (LPS)-induced inflammation of dental pulp cells [120]. m6A-methylated mRNA was also involved in nuclear export [121], which is supported via the disruption of the nuclear export of methylated mRNA via the WTAP knockdown [122]. Meanwhile, m6A methylation could recruit nuclear export factors, such as nuclear RNA export factor 1 (NXF1) and nuclear transport factor 2 like export factor 1 (NXT1) with mRNA to promote nuclear export [123]. The silencing of *Mettl3* suppressed the pre-mRNA export of the circadian clock transcript, resulting in a prolonged circadian rhythm [124]. In the process of protein production, m6A methylation mediated translation regulation. METTL3 could recruit a subunit of the translation initiation factor eIF3h to the 3′UTR to promote mRNA translation, and the corresponding deletion of *Mettl3* reduced mRNA translation [125,126]. In addition, the interaction of METTL3 with the transcription factor CCAAT enhancer-binding protein zeta (CEBPZ) in the promoter could also promote mRNA translation [127]. On the contrary, m6A methylation could negatively affect protein production. Several studies have found that the presence of m6A methylation in the CDS of oocytes was associated with reduced protein levels [128,129]. The silencing of METTL3 in mammalian embryonic cells induced a longer half-life of mRNA transcripts, suggesting that m6A methylation was involved in the stability of mRNA [87,130].

A large numbers of studies have revealed that m6A modification plays a role in brain functions such as synaptic plasticity, learning, and memory [62,64,65,66,105,131]. Collectively, the loss of m6A function led to deficits in synaptic plasticity, learning, and memory in mice. Synaptic plasticity is the fundamental structural basis of neuronal network and function. The basis of learning and memory activities depends on changes in the strength of synaptic connections, and one underlying mechanism for the continuous change in synaptic strength depends on the synthesis of new proteins. The synthesis of new proteins requires gene expression and a series of potential mRNA modifications and regulations. Several studies of interference with m6A-related enzymes have revealed potential roles for m6A in brain function. The knockdown of m6A readers (YTHDF1 or YTHDF3) by shRNA in hippocampal neurons led to synaptic dysfunction [132]. The knockout of *Mettl3* in postnatal mice resulted in the loss of m6A and an insufficient translation of immediate early genes, revealing the association of METTL3 with learning and long-term memory [65]. The deletion of *Mettl14* in the striatum led to decreased m6A levels, which further impaired striatum-mediated learning and performance and increased neuronal excitability [64]. FTO plays a role in learning and memory. The FTO knockdown in the dorsal hippocampus in mice enhanced contextual fear memory [133]. The knockout of *Ythdf1* led to reduced dendritic spines density in CA1 neurons and learning and memory deficits, revealing that YTHDF1 could promote the translation of m6A-modified mRNAs, thereby promote learning and memory [134]. Furthermore, m6A also played a key role in cognition [66,135]. In humans and mice, global m6A dynamically exhibited temporal and spatial signatures that accumulated during development and maturation [1,136]. Aging is well known to be a significant factor in cognitive decline. The reduced m6A levels in aged mice have been found to lead to a decrease in synaptic protein calcium/calmodulin dependent protein kinase II (CAMK II) and glutamate ionotropic receptor AMPA type subunit 1 (GLUA1) synthesis during cognitive decline [131]. In addition, m6A modification is also involved in the regulation of senescence and apoptosis [137,138].

Myelination is an important physiological process in the brain and serves as the structural basis for neuronal function. Oligodendrocytes are the main players in myelination. The differentiation and maturation process of oligodendrocytes are regulated via m6A modification [139,140]. Proline-rich coiled-coil 2 A (*Prcc2a*) has been shown to regulate oligodendrocyte transcription factor 2 (*Olig2*) mRNA stability as an m6A reader, and *Prcc2a* knockout mice led to motor and cognitive impairments, demyelination, and a shortened lifespan [141]. In addition, 6mA modification was altered in LPS-stimulated microglia [142], and m6A reader Igf2bp1 enhanced the inflammatory response of microglia by enhancing Guanylate-binding protein 11 (*Gbp11*) and Ceruloplasmin (*Cp*) mRNA stability [143], highlighting a regulatory role for m6A in neuroinflammation. In summary, these studies demonstrated that m6A, as an epitranscriptomic modification of RNA, was involved in synaptic function, learning and memory, myelination, and neuroinflammation of the brain (Table 1).

Similar to other epigenetic modifications, m6A bridges the gap between the regulation of gene expression and environmental factors, including ultraviolet or visible light, diet, exercise, air pollutants, and heavy metal exposure. Ultraviolet B (UVB) exposure downregulated METTL14 and induced skin tumorigenesis in humans and mice [163]. METTL3 regulated melanin synthesis and DNA repair systems via its methyltransferase activity during UVB exposure [164,165]. It has been shown that continuous light exposure may affect m6A dynamics and/or other epigenetic modifications, thereby interfering with fertility and cognition in mice [101,166]. Particularly, chronic blue light exposure during sleep affected the m6A modification and mRNAs expression associated with the calcium signaling pathways and long-term depression [167]. Lifestyle is an important factor affecting the m6A epitranscriptome, including diet [168,169,170,171], fasting [172], exercise [173,174,175,176], and cigarette exposure [177,178,179,180,181], which may affect physiological function and disease risk. A prominent component of air pollutants is particulate matter 2.5, which can cause serious damage to the respiratory system after overexposure [182,183]. The possible reason is that the exposure of particulate matter 2.5 interfered with the expression and function of METTL3, METTL14, METTL16, ALKBH5, YTHDF1, and YTHDF2, which in turn led to the dysregulation of the epitranscriptome [184,185,186,187,188,189,190,191,192]. Interestingly, RNA m6A modification also mediated microplastics-induced cardiac damage [193,194]. Arsenite is an environmental pollutant that can cause cellular stress. Arsenite exposure may induce the abnormal expression of m6A writers and erasers, affect the regulatory function of m6A, and cause a stress response [195,196,197,198]. In addition, heavy metals are important environmental factors related to human diseases, including cadmium, cobalt, manganese, lead, etc. Emerging evidence elucidated potential mechanisms by which cadmium (Cd) [199,200,201,202,203], cobalt (Co) [204,205,206,207], manganese (Mn) [100,208], and lead (Pb) [209,210] affected RNA m6A modification and disease initiation and progression, respectively. Overall, m6A modification is involved in a variety of environmental factors mediated via physiological dysfunction or disease, but it is also a potential epigenetic intervention in the aforementioned diseases.

## 6. RNA m6A in Neurodegenerative Diseases

Neurodegenerative diseases are marked via the progressive loss of neurons in the brain and/or spinal cord. Typical neurodegenerative diseases include Alzheimer’s disease (AD) and Parkinson’s disease (PD). Numerous studies have shown that epigenetic modifications, such as 5mC, 5hmC, and m6A, are involved in the occurrence and development of neurodegenerative diseases. Here, we summarized the recent progress of potential roles of m6A in AD and PD to provide novel therapeutic candidates.

AD usually occurs in people over the age of 65 and is characterized by the cerebral cortical atrophy, β-amyloid deposits, neurofibrillary tangles, the impairment of memory neurons, and senile plaques [211]. So far, the mutations in the amyloid protein precursor (APP), presenilin-1 (PSEN1), and presenilin-2 (PSEN2) genes have been identified in the pathogenesis of AD. Amyloid deposition led to tangle formation, neuroinflammation, synaptic dysfunction, and neuronal loss [212]. The alteration in synaptic function in AD patients is more important. m6A modification and the related proteins are dysregulated and abnormally expressed in the brain of AD patients [144]. Zhao et al. revealed the m6A changes in different brain regions/cells of AD patients, as well as the reduced expression of the writers (METTL3, METTL14, and WTAP), eraser (FTO), and reader (YTHDF1), and also revealed abnormality nuclear localization of METTL3/14 [145]. Furthermore, the expression changes of m6A regulators FTO, ELAVL1, and YTHDF2 in AD brains have been found, which affected memory and cognition [146], suggesting a link between m6A dynamics and AD development.

One of the obvious markers in AD patients is the highly phosphorylated tau protein. The hyperphosphorylated tau protein forms intracellular aggregates called Neurofibrillary tangles (NFTs). The latest research showed that the Lentivirus-mediated FTO knockdown reduced the phosphorylation level of the tau protein. Conversely, FTO overexpression increased the phosphorylation level of neuronal tau [147]. The main mechanism was that FTO activated the mammalian target of rapamycin (mTOR) and its downstream signaling pathways, which activated the phosphorylation of tau in a mTOR-dependent manner. Meanwhile, the knockout of FTO also reduced cognitive deficits in AD mice [147]. HNRNPA2B1 acted as an m6A reader in the cytoplasm, linking m6A RNA to oligomeric tau in pathological conditions, resulting in neurotoxicity [148]. In addition, upregulated METTL3 promoted the autophagic clearance of phosphorylated tau, playing a protective role [149]. Collectively, m6A modification and its interacting proteins regulated RNA metabolism in the CNS and were involved in the tau pathology of AD.

m6A and its related molecules also affect the β-amyloid(Aβ) pathology during the AD process. Compared with wild-type mice, there were differences in circRNA m6A methylation and the gene expression levels in the brain of AD mice, which might be involved in AD progression [213]. Dynamic analyses of m6A differences in these AD model mice revealed abnormalities of 6mA AD development [214]. The activity-regulated cytoskeleton-associated protein (ARC) was decreased in the brains of AD patients, resulting in impaired memory consolidation. However, Aβ inhibited RNA m6A modification, whereas METTL3 activated the ARC expression in a METTL3-m6A-YTHDF1-dependent manner, thereby rescuing AD-induced ARC downregulation [150]. Another study showed that the deletion of METTL3 downregulated α-tubulin acetyltransferase 1 (ATAT1), reduced α-tubulin, and promoted Aβ clearance [151]. The m6A reader protein insulin-like growth factor 2 mRNA-binding protein 2 (IGF2BP2) was associated with extracellular matrix–receptor (ECM receptor) interaction, focal adhesion, cytokine–cytokine receptor interaction, and the TGF-β signaling pathways, and participated in the occurrence of AD [152]. In addition, NADH-ubiquinone oxidoreductase subunit A10 (NDUFA10) may be regulated via m6A modification, thereby affecting energy metabolism in AD [153].

PD is the second most common neurodegenerative disease, attributed to the loss of dopaminergic neurons in the substantia nigra and the extensive accumulation of intracellular α-synuclein (SNCA). Typical symptoms in PD patients include a resting tremor, bradykinesia, and rigidity [215,216]. The excessive loss of dopaminergic neurons occurs before the onset of motor symptoms in PD patients [217]. m6A modification variants were found in PD patients [218], and METTL3, METTL14, and YTHDF2 mRNA levels were lower in PD patients, suggesting that PD pathogenesis may be regulated via m6A modification [154].

Mn is one of the important environmental factors for the pathogenesis of PD. Mn caused dopaminergic neuron projection damage and movement disorders via the Foxo3a/FTO/m6A/ePhin-B2/YTHDF2 signaling pathway [100]. FTO deficiency in mice resulted in the increased methylation and expression levels of mRNAs associated with the dopamine (DA) signaling pathways in the midbrain and striatum, revealing a correlation between FTO mRNA regulation and DA transmission function [155]. Interestingly, when FTO was overexpressed or treated with m6A inhibitors, the reduction in m6A resulted in the decreased expression of N-methyl-D-aspartate (NMDA) receptor 1 and increased oxidative stress, Ca^2+^ influx, and the apoptosis of dopaminergic neurons [156]. The protective effect of the small molecule inhibitor targeting FTO on dopaminergic neurons demonstrated the important role of FTO in the pathology of PD and the therapeutic potential of its inhibitors [157]. Although many studies have reported the role of FTO in dopaminergic neurons in PD models, the specific mechanism has not been fully elucidated. The conditional knockout of METTL14 significantly reduced the overall mRNA m6A levels and downregulated Nurr1, pitx3, and EN1 expressions in the substantia nigra region, which was associated with behavioral motor impairment [158]. Furthermore, METTL14 regulates the stability of α-syn mRNA [154]. The HNRNPC expression level is low in PC12 cells, while HNRNPC overexpression inhibits apoptosis and immune inflammation [159]. In the 1-methyl-4-phenyl-1,2,3,6-tetrahydropyridine (MPTP)-induced PD mouse model, m6A-related proteins exhibited differential expression levels, such as ALKBH5 and YTHDF1, suggesting the potential role of m6A modification in PD pathology [160].

Recent published data have also described the association between m6A modification and neurodegenerative diseases, including amyotrophic lateral sclerosis (ALS) and Huntington’s disease (HD). ALS is a progressive neurodegenerative disease characterized by the degeneration of motor neurons in the brain and spinal cord. An abnormal cytoplasmic accumulation of TDP-43 (TAR-DNA-binding protein of 43 kDa) has been observed in up to 95% of ALS patients. TDP-43 is preferentially bound to m6A-modified mRNA, and the knockdown of YTHDF2 in spinal cord neurons carrying ALS-associated mutations attenuated TDP43-associated neurotoxicity and prolonged survival [162]. HD is a rare and inherited neurodegenerative disease characterized by involuntary dance-like movements of the body, anxiety, depression, apathy, and irritability. The cause of HD is the mutation of the Huntingtin gene on the fourth chromosome, resulting in the mutated Huntington protein (mHTT). The knockdown of FTO in the hippocampal CA1 region of the HD mouse model (*Hdh*+/*Q111* mouse) improved hippocampal spatial and recognition memories [162], demonstrating an underlying role for m6A regulation in hippocampal memory function in HD.

Despite numerous studies reporting potential roles of m6A-dependent RNA regulation and the associated proteins, many unknown mechanisms need to be elucidated. So far, there is no report on the role of DNA 6mA in the pathogenesis of neurodegenerative diseases. As new epigenetic modifications, the potential functions of DNA 6mA and RNA 6mA in neurodegenerative diseases will be a prospective and challenging subject. Further understanding of the specific regulatory mechanisms will provide new opportunities for future clinical treatment.

## 7. Conclusions and Outlook

With the rapid development of high-throughput sequencing and transcriptomic analysis, the precise localization of DNA 6mA and RNA m6A methylation in the entire genome and transcriptome has enabled us to gain a deeper understanding of the processes involved in 6mA and m6A in eukaryotes. Current studies have highlighted the potential functions of writers, erasers, and readers of 6mA and m6A in various biological and pathological processes.

In mammals, the proportion of m6A modification is approximately 0.1–0.4%, with each mRNA containing 3–5 m6A modification sites. On the other hand, there are fewer than 10 6mA modification sites per million adenines in DNA. Methylation, achieved via the addition of methyl groups to DNA or RNA adenines, ultimately influences translation by differentially regulating target gene expression. Notably, m6A, being the most abundant modification in mRNA, plays a crucial role in mRNA splicing, translation, and localization. Moreover, the abnormal expression of m6A methylation-related enzymes in the mammalian brain has significant implications for brain development, memory, neurons, and the pathogenesis of neurodegenerative diseases. However, research on the identification and function of 6mA is still in its preliminary stages. Although 6mA and m6A methylation share the same chemical process, their consequences in organisms are distinct. Therefore, it is worth discussing the relationship between 6mA and m6A modifications, as well as their targets and regulatory functions.

In conclusion, DNA 6mA and RNA m6A serve as novel epigenetic markers in the regulation of gene expression and biological functions. The application of advanced technologies, such as CRISPR/Cas9, holds great significance for analyzing the intricate regulation of 6mA and m6A, as well as their associated enzymes, in a wider range of species. Further extensive investigations will provide comprehensive insights into the molecular mechanisms of DNA 6mA and RNA m6A in the mammalian genome.

## Figures and Tables

**Figure 1 ijms-24-13897-f001:**
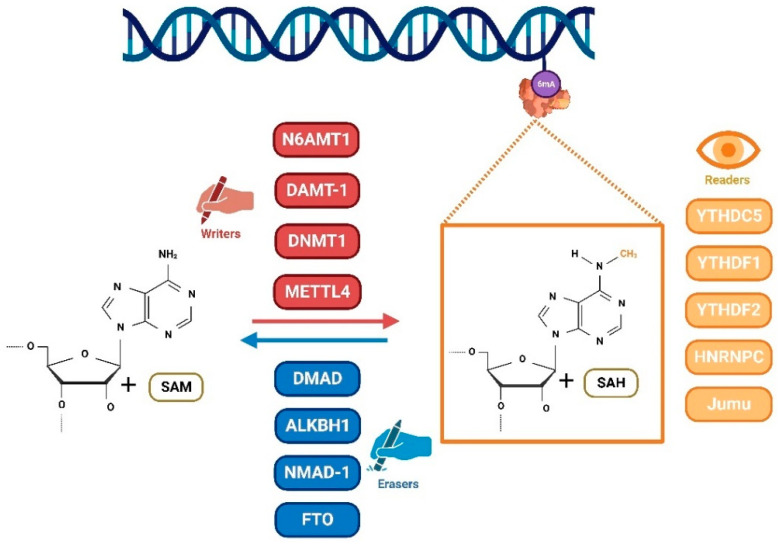
DNA 6mA modification is a dynamic process. Methyltransferases (writers), including N6AMT1, DAMT-1, DNMT1, and METTL4 add a methyl group to form 6mA. This process can be reversed via demethylases (erasers) such as DMAD, ALKBH1, NMAD-1, and FTO. In addition, binding proteins (readers) recognize and bind the 6mA site on DNA. Therefore, they also play an important role in the regulation of gene expression and cellular functions.

**Figure 2 ijms-24-13897-f002:**
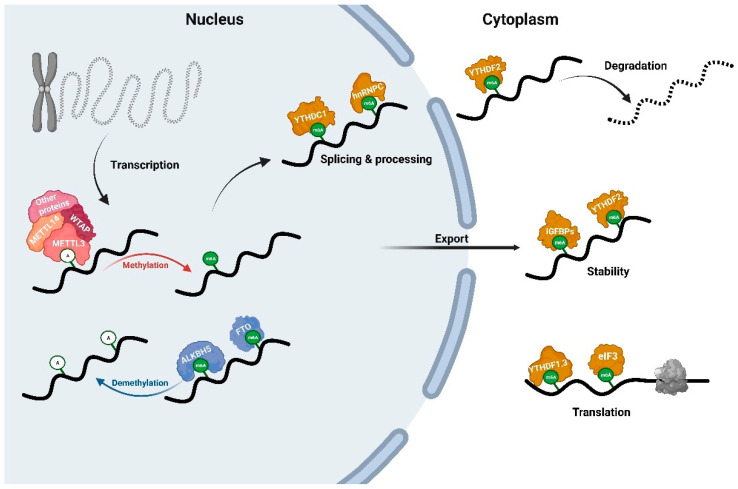
RNA m6A dynamics and mRNA regulation. RNA m6A modification is generated by the writers complex genes (METTL3, METTL14, WTAP, and other proteins) and reversed by the erasers (ALKBH5 and FTO) in the nucleus, which perform mRNA regulatory functions in the nucleus and cytoplasm, including splicing, processing, degradation, stability, and translation.

**Table 1 ijms-24-13897-t001:** Association of m6A modifications with neurodegenerative diseases.

Models	Regulators	m6A Level	Phenotype	References
Aging	METTL3 ↑	Increase	METTL3 expression and global m6A levels increase during aging in mice and human.	[59,72]
	METTL3 ↓	Decrease	METTL3 reduction decreases NPNT expression and m6A levels in human pluripotent stem cell-derived myotubes, leading to senescence and apoptosis.	[138]
	METTL3 KD	Decrease	Accelerates hMSC senescence.	[137]
	METTL3 OE	Increase	Rescues hMSC senescence.	
AD	METTL3 ↓, RBM15B ↑	-	Downregulation of METTL3 and upregulation of RBM15 were detected in AD human hippocampus.	[144]
	METTL3 ↓, METTL14 ↓, WTAP ↓, FTO ↓, YTHDF1 ↓	Decrease	Reduces expression of METTL3, METTL14, FTO, and YTHDF1, and reduces m6A levels and abnormality of METTL3/14 nuclear localization in AD patient’s pyramidal neurons.	[145]
	METTL3 KD	Decrease	Promotes neuronal death in the hippocampus, Aβ oligomer induced cognitive and memory impairments in mice.	
	METTL3 OE	Increase	Rescues the effects of METTL3 KD.	
	RBM15 ↓, FTO ↓, ELAVL1 ↓, YTHDF2 ↓	-	Downregulation of RBM15 and FTO and upregulation of ELAVL1 and YTHDF2 in AD hippocampus affects memory and cognition.	[146]
	FTO ↓	Increase	Reduces the phosphorylation level of tau protein in AD mice (may decrease *Tsc1* mRNA m6A level and its stability).	[147]
	FTO OE	Decrease	Rescues the effects of FTO KD.	
	HNRNPA2B ↓	-	Reduces oligomeric tau-induced neurotoxicity in mice.	[148]
	METTL3 ↑	Increase	Promotes autophagic clearance of phosphorylated tau.	[149]
	METTL3 ↓	Decrease	Reduces ARC protein expression.	[150]
	METTL3 ↑	Increase	Rescues the effects of METTL3 knockdown.	
	METTL3 KD	Decrease	Downregulates ATAT1, decreases α-tubulin, and promotes Aβ clearance.	[151]
	IGF2BP2 ↑	-	IGF2BP2 shows an increased level in AD patients.	[152]
	METTL3 ↓	Decrease	Reduction in METTL3 leads to decreased expression of NDUFA10, affecting electronic respiratory chain function.	[153]
PD	METTL3 ↓, METTL14 ↓, YTHDF2 ↓	Decrease	METTL3, METTL14, YTHDF2, and m6A levels were lower in PD patients.	[154]
	METTL14 OE	Increase	Increases α-synuclein mRNA m6A level and decreases its stability.	
	FTO ↓	Increase	Mn decreases FTO level and causes dopaminergic neuron projection damage and movement disorders.	[100]
	FTO OE	Decrease	FTO and ephrin-B2 overexpression increases survival rate of neurons.	
	FTO ↓	Increase	Increased methylation of mRNAs associated with DA signaling pathways in the midbrain and striatum.	[155]
	FTO ↑	Decrease	Decreases expression of N-methyl-D-aspartate (NMDA) receptor 1, increases oxidative stress, Ca^2+^ influx, and apoptosis of dopaminergic neurons.	[156]
	FTO ↓	Increase	Small molecule inhibitor targeting FTO promotes survival of dopaminergic neurons.	[157]
	METTL14 cKO	Decrease	Downregulates *Nurr1*, *pitx3*, and *EN1* expression in the substantia nigra region.	[158]
	HNRNPC OE	-	Inhibits apoptosis and immune inflammation in PC2 cells.	[159]
	ALKBH5 ↑, IGF2BP2 ↑, YTHDF1↓, METTL3↓, FMR ↑, CBLL1 ↑, RBM15 ↓	Decrease	m6A-related proteins showed differential expression levels in SN and striatum in MPTP-induced PD mouse model.	[160]
ALS	YTHDF2 KD	-	YTHDF2 knockdown attenuates TDP43-associated neurotoxicity and prolongs survival.	[161]
HD	FTO KD	Increase	FTO knockdown in the hippocampal CA1 region of HD mouse model (*Hdh*+/*Q111* mouse) improved hippocampal spatial and recognition memories	[162]

(↑: upregulation; ↓: downregulation; KD: knockdown; cKO: conditional knockout; OE: overexpression).

## Data Availability

Not applicable.

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
