# Peer review of "Emerging Roles for DNA 6mA and RNA m6A Methylation in Mammalian Genome"

_ijms, 2023, doi:10.3390/ijms241813897_

Round 1

Reviewer 1 Report

This review by Xie et al. aims to summarize and compare what is known about DNA-6mA and RNA-m6A methylation in the mammalian genome. It describes the writers, erasers, and readers of each and their potential functions. It goes on to discuss only the role of RNA-m6A in neurodegenerative diseases. The reason for this section is somewhat unclear because the rest of the manuscript is a comparison, and even though little is known about the function of DNA-6mA, there is no attempt to tie in how the RNA-m6A data might be extrapolated to a role for DNA-6mA in neurodegenerative diseases. Without any kind of tie in, the neurodegenerative diseases section seems out of place. Otherwise, the manuscript is informative, well-written, and not a topic overly reviewed in the literature. I recommend publication with the following minor comments.

1. Provide some kind of relevance for discussing the role of RNA-m6A in neurodegenerative diseases. What is the reason this was included in the review? Perhaps, conclude this section bringing back the fact that DNA-6mA levels are highest in the brain and speculating why DNA-6mA could also play a role in neurodegenerative diseases and propose some studies that need to be done to look at this potential.

2. Figure 1 lists the writers and erasers of DNA-6mA, but not the readers, these could also be listed near the “Reader” icon

Author Response

This review by Xie et al. aims to summarize and compare what is known about DNA-6mA and RNA-m6A methylation in the mammalian genome. It describes the writers, erasers, and readers of each and their potential functions. It goes on to discuss only the role of RNA-m6A in neurodegenerative diseases. The reason for this section is somewhat unclear because the rest of the manuscript is a comparison, and even though little is known about the function of DNA-6mA, there is no attempt to tie in how the RNA-m6A data might be extrapolated to a role for DNA-6mA in neurodegenerative diseases. Without any kind of tie in, the neurodegenerative diseases section seems out of place. Otherwise, the manuscript is informative, well-written, and not a topic overly reviewed in the literature. I recommend publication with the following minor comments.

Response: We thank the reviewer for the affirmation of the manuscript.

We have addressed the reviewer’s concerns point-by-point as follows:

  1. Provide some kind of relevance for discussing the role of RNA-m6A in neurodegenerative diseases. What is the reason this was included in the review? Perhaps, conclude this section bringing back the fact that DNA-6mA levels are highest in the brain and speculating why DNA-6mA could also play a role in neurodegenerative diseases and propose some studies that need to be done to look at this potential.

Response: Over the past few years, we have focused on dynamic changes of DNA methylation during aging and neurodegenerative diseases. Our study found that monkey brain has extremely high level of 5hmC which accumulate with aging (Frontiers in Aging Neuroscience, 2022). The reason we focus on neurodegenerative diseases is due to the higher abundance of 6mA and m6A in the mammalian brain than in other tissues. Importantly, our unpublished data showed 6mA accumulation in the brain of monkeys with aging, suggesting an epigenetic regulatory role of 6mA in normal aging. We have added an explanation of why the potential role of m6A in neurodegenerative diseases is discussed on Page 2 line 56-65.

  1. Figure 1 lists the writers and erasers of DNA-6mA, but not the readers, these could also be listed near the “Reader” icon.

Response: We have replaced Figure 1 by adding the readers and Figure 2 with high resolution.

Reviewer 2 Report

Page no. 2 line 25. Author should write DNA modification, instead of DNA methylation as DNA methylation is not the only epigenetic modification.

page No. 2 line 39. Author stated that "A currently controversial......" Author should cite the statement. 

Page no. 4 line 117. Author should provide some details of Jamu protein and Zelda gene what is their role and importance.

Author should give some details of each mentioned gene regarding their role and and importance in cellular homeostasis and disease pathology.

Author should give insight into epigenetic modification mechanism for each mentioned gene, in which type of condition there modification occurs.

Kumar et 2021, Environmental Research, reported that epigenetic modification in rat brain hippocampus under the influence of environmental factors mobile phone signal radiation. Author should include a section for environmental factors in epigenetic modulation in different model.

Author Response

  1. Page no. 2 line 25. Author should write DNA modification, instead of DNA methylation as DNA methylation is not the only epigenetic modification.

Response: We have revised it.

  1. page No. 2 line 39. Author stated that "A currently controversial......" Author should cite the statement. 

Response: We have added three citations here.

  1. Page no. 4 line 117. Author should provide some details of Jamu protein and Zelda gene what is their role and importance.

Response: We supplemented the ‘6mA reader’ section and described the functions of the Jumu and Zelda proteins in detail. Please check Page 3, line 123-131.

  1. Author should give some details of each mentioned gene regarding their role and importance in cellular homeostasis and disease pathology.

Response: We have added details of mentioned genes in the text on Page 3, line 117-119, 123-131. For genes with defined functions, we have described; for genes with unclear functions, we have discussed their potential changes and roles.

  1. Author should give insight into epigenetic modification mechanism for each mentioned gene, in which type of condition there modification occurs.

Response: Epigenetic modifications occur throughout life, including germ cell formation, embryogenesis, development, and aging. The key genes either continue to play a regulatory role, or paly a regulatory role in a specific stage. Meanwhile, the environmental factors contributed to the epigenetic changes. This is complex and coordinated process caused by various factors. It is difficult to define one or two specific conditions under which modification occurs.

  1. Kumar et 2021, Environmental Research, reported that epigenetic modification in rat brain hippocampus under the influence of environmental factors mobile phone signal radiation. Author should include a section for environmental factors in epigenetic modulation in different model.

Response: We thank the reviewer for the constructive suggestion. We have included a section about the environmental stress and 6mA changes on Page 5, line 198-215; and another section about the environmental stress and m6A changes on Page 11, line 417-443.

Round 2

Reviewer 1 Report

My comments have been addressed.

Reviewer 2 Report

Author answers and justified all raised questions and queries.